# CERTIFIED ROBUSTNESS TO ADVERSARIAL LABEL-FLIPPING ATTACKS VIA RANDOMIZED SMOOTHING

## ABSTRACT

This paper considers label-flipping attacks, a type of data poisoning attack where an adversary relabels a small number of examples in a training set in order to degrade the performance of the resulting classifier. In this work, we propose a strategy to build linear classifiers based on deep features that are *certifiably* robust against a strong variant of label-flipping, where the adversary can target each test example independently. In other words, for each test point, our classifier makes a prediction and includes a certification that its prediction would be the same had some number of training labels been changed adversarially. Our approach leverages randomized smoothing, a technique that has previously been used to guarantee test-time robustness to adversarial manipulation of the input to a classifier. Further, we obtain these certified bounds with *no additional runtime cost* over standard classification. On the Dogfish binary classification task from ImageNet, in the face of an adversary who is allowed to flip 10 labels to individually target each test point, the baseline undefended classifier achieves no more than 29.3% accuracy; we obtain a classifier that maintains 64.2% certified accuracy against the same adversary. We generalize our results to the multi-class case, providing what we believe to be the first multi-class classification algorithm that is certifiably robust to label-flipping attacks.

## 1 INTRODUCTION

Modern classifiers, despite their widespread empirical success, are known to be susceptible to adversarial attacks. In this paper, we are specifically concerned with so-called "data-poisoning" attacks (formally, *causative* attacks [Barreno et al. 2006; Papernot et al. 2018]), where the attacker manipulates some aspects of the training data in order to cause the learning algorithm to output a faulty classifier. Work in this area includes label-flipping attacks (Xiao et al., 2012), where the labels of a training set can be adversarially manipulated to decrease performance of the trained classifier; general data poisoning, where both the training inputs and labels can be manipulated (Steinhardt et al., 2017); and backdoor attacks (Chen et al., 2017; Tran et al., 2018), where the training set is corrupted so as to cause the classifier to deviate from its expected behavior when triggered by a specific pattern, without causing a noticeable drop in overall accuracy. However, unlike the alternative "test-time" adversarial setting, where reasonably effective defenses exist to build adversarially robust classifiers, relatively little work has been done on building classifiers that are certifiably robust to targeted data poisoning attacks.

In this work, we propose a strategy for building classifiers that are certifiably robust against label-flipping attacks. In particular, we propose a *pointwise* certified defense—by this we mean that with each prediction, the classifier includes a certification guaranteeing that its prediction would not be different had it been trained on data with some number of labels flipped. Existing works on certified defenses make statistical guarantees over the entire test set, but they make no guarantees as to the robustness of a prediction on any particular test point. Thus, while these algorithms provide general robustness to any single corruption, a determined adversary could still cause a specific test point to be misclassified. We therefore consider the threat of a worst-case adversary that can make a training set perturbation to specifically target *each test point individually*. This motivates a defense that can certify each of its individual predictions, as we present here. To the best of our knowledge, this work represents the first pointwise certified defense to data poisoning attacks.

Our approach leverages randomized smoothing (Cohen et al., 2019), a technique that has previously been used to guarantee test-time robustness to adversarial manipulation of the input to a deep network. However, where prior uses of randomized smoothing randomize over the input to the classifier for test-time guarantees, we instead randomize over *the entire training procedure of the classifier*. Specifically, by randomizing over the labels during this training process, we obtain an overall classification pipeline that is certified to be robust (i.e., to not change its prediction) when some number of labels are adversarially manipulated in the training set. Although a naive implementation of this approach would not be computationally feasible, we show that by using a linear least-squares classifier over the output of a pre-trained network, we can obtain these certified bounds with *no additional runtime cost* over standard classification—and we again emphasize that this is the first such certified defense against pointwise label-flipping attacks, even for linear classifiers.

We evaluate our proposed classifier on several benchmark datasets common to the data poisoning literature. Specifically, we demonstrate that our randomized classifier is able to achieve 75.7% certified accuracy on MNIST 1/7 even when the number of allowed label flips would drive a standard, undefended classifier to less than 50%. Similar results in experiments on the Dogfish binary classification challenge from ImageNet and IMDB review sentiment database validate our technique for more challenging datasets, and we further experiment on the full MNIST dataset to demonstrate its effectiveness for multi-class classification. On top of this, our classifier maintains a reasonably competitive non-robust accuracy (e.g., 94.5% on MNIST compared to 99.1% for the undefended classifier).

## 2    RELATED WORK

**Data-poisoning attacks**    A *data-poisoning attack* (Muñoz González et al., 2017; Yang et al., 2017) is an attack whereby an adversary corrupts some portion of a training set or adds new inputs, with the goal of degrading the performance of the learned model. The attack can be targeted to cause poor performance on a specific test example or can instead simply reduce the overall test performance. The adversary is assumed to have perfect knowledge of the learning algorithm, so security by *design*—as opposed to obscurity—is the only viable defense against such attacks. The adversary is also typically assumed to have access to the training set and, in some cases, the test set.

Previous work has investigated attacks and defenses for data-poisoning attacks applied to feature selection (Xiao et al., 2015), SVMs (Biggio et al., 2011; Xiao et al., 2012), linear regression (Liu et al., 2017), and PCA (Rubinstein et al., 2009), to name a few. Some attacks can even achieve success with "clean-label" attacks, inserting innocuous, poisoned yet "correctly" labeled training examples but causing the classifier to perform poorly (Shafahi et al., 2018; Zhu et al., 2019). For an overview of data poisoning attacks and defenses in machine learning, see Biggio et al. (2014).

**Label-flipping attacks**    A *label-flipping attack* is a specific type of data-poisoning attack where the adversary is restricted to changing some of the training labels. The number of allowed changes could be an exact number or a percentage of the total labels. The classifier is then trained on the corrupted training set, with no knowledge of which labels have been tampered with.

Unlike random label noise, for which many robust learning algorithms have been successfully developed (Natarajan et al., 2013; Liu & Tao, 2016; Patrini et al., 2017), adversarial label-flipping attacks can be specifically targeted to exploit the structure of the learning algorithm, significantly degrading performance. Robustness to such attacks is therefore harder to achieve, both theoretically and empirically (Xiao et al., 2012; Biggio et al., 2011). A common defense technique is *sanitization*, whereby a defender attempts to identify and remove or relabel training points that may have had their labels corrupted (Paudice et al., 2019; Taheri et al., 2019). Unfortunately, recent work has demonstrated that this is often not enough against a sufficiently powerful adversary (Koh et al., 2018).

**Certified defenses**    For a review on robustness to random noise, we refer the reader to Rousseeuw & Leroy (1987)—our focus is on adversarial noise, a much more challenging problem. Existing works on certified defenses to adversarial data poisoning attacks typically focus on the regression case and provide broad statistical guarantees over the entire test distribution. A common approach to such certifications is to show that a particular algorithm recovers some close approximation to the best linear fit coefficients (Liu et al., 2017; Prasad et al., 2018; Shen & Sanghavi, 2019), or that the

expected loss on the test distribution is bounded (Klivans et al., 2018; Chen & Paschalidis, 2018). These results generally rely on assumptions on the train and test distributions: some assume sparsity in the coefficients (Karmalkar & Price, 2018; Chen et al., 2013) or corruption vector (Bhatia et al., 2015); others require limited effects of outliers (Steinhardt et al., 2017). As mentioned above, all of these methods fail to provide guarantees for individual test points. Additionally, most of these statistical guarantees are not meaningful when applied to deep, non-linear architectures.

**Randomized smoothing** Since the discovery of adversarial examples (Szegedy et al., 2013; Goodfellow et al., 2015), the research community has been investigating techniques for increasing the robustness of deep networks to adversarial perturbations. After a series of heuristic defenses (Metzen et al., 2017; Feinman et al., 2017), followed by attacks breaking them (Athalye et al., 2018; Carlini & Wagner, 2017), focus began to shift towards the development of *provable* robustness.

One approach which has gained popularity in recent work is randomized smoothing. Rather than certifying the original classifier $f$, randomized smoothing defines a new classifier $g$ whose prediction at an input $x$ is the class assigned the most probability when $x$ is perturbed with noise from some distribution $\mu$ and passed through $f$. That is, $g(x) = \arg\max_c \mathbb{P}_{\epsilon \sim \mu}(f(x + \epsilon) = c)$. This new classifier $g$ is then certified as robust, ideally without sacrificing too much accuracy compared to $f$. The original formulation was presented by Lecuyer et al. (2018) and borrowed ideas from differential privacy. The above definition is due to Li et al. (2018) and was popularized by Cohen et al. (2019), who derived a tight robustness guarantee. Follow-up work has focused on optimizing the training procedure of $f$ (Salman et al., 2019) and extending the analysis to other types of distributions (Lee et al., 2019). For more details, we refer the reader to Cohen et al. (2019).

## 3 A GENERAL VIEW OF RANDOMIZED SMOOTHING

We begin by presenting a general viewpoint of randomized smoothing. Under our notation, randomized smoothing constructs an operator $G(\mu, \phi)$ that maps a binary-valued [1] function $\phi : \mathcal{X} \to \{0, 1\}$ and a *smoothing measure* $\mu : \mathcal{X} \to \mathbb{R}_+$, with $\int_{\mathcal{X}} \mu(x)dx = 1$, to the expected value of $\phi$ under $\mu$ (that is, $G$ represents the "vote" of $\phi$ weighted by $\mu$). For example, $\phi$ could be a binary image classifier and $\mu$ could be some small, random pixel noise applied to the to-be-classified image. We also define a "hard threshold" version $g(\mu, \phi)$ that returns the most probable output (the majority vote winner). Formally,

$$G(\mu, \phi) = \mathbf{E}_{x \sim \mu}[\phi(x)] = \int_{\mathcal{X}} \mu(x)\phi(x)dx \quad \text{and} \quad g(\mu, \phi) = \mathbf{1}\{G(\mu, \phi) \geq 1/2\}, \quad (1)$$

where $\mathbf{1}\{\cdot\}$ is the indicator function. Intuitively, for two smoothing measures $\mu, \rho : \mathcal{X} \to \mathbb{R}_+$ that are very similar, we would expect that for most $\phi$, even though $G(\mu, \phi)$ and $G(\rho, \phi)$ may not be equal, the threshold function $g$ should satisfy $g(\mu, \phi) = g(\rho, \phi)$. Further, the degree to which $\mu$ and $\rho$ can differ while still preserving this property should increase as $G(\mu, \phi)$ approaches either 0 or 1, because this increases the "margin" with which the function $\phi$ is 0 or 1 respectively over the measure $\mu$. More formally, we define a general randomized smoothing guarantee as follows.

**Definition 1.** *Let $\mu : \mathcal{X} \to \mathbb{R}_+$ be a smoothing measure over $\mathcal{X}$, with $\int_{\mathcal{X}} \mu(x)dx = 1$. Then a randomized smoothing robustness guarantee is a specification of a distance measure $d(\mu, \rho)$ and a function $f : [0, 1] \to \mathbb{R}_+$ such that for all $\phi : \mathcal{X} \to \{0, 1\}$ and all measures $\rho : \mathcal{X} \to \mathbb{R}_+$ of a given form,*

$$g(\rho, \phi) = g(\mu, \phi) \quad \text{whenever} \quad d(\mu, \rho) \leq f(G(\mu, \phi)). \quad (2)$$

For brevity, we will sometimes use $p$ in place of $G(\mu, \phi)$, representing the probability with which the majority class wins the vote (this is analogous to $p_A$ in Cohen et al. (2019)).

**Instantiations of randomized smoothing** This definition is rather abstract, so we highlight concrete examples of how it can be applied to achieve certified guarantees against adversarial attacks.

---

[1]For simplicity, we present the methodology here in terms of binary-valued functions, which will correspond eventually to binary classification problems. The extension to the multiclass setting requires additional notation, and thus is deferred to the appendix.

**Example 1.** *The randomized smoothing guarantee of Cohen et al. (2019) uses the smoothing measures $\mu = \mathcal{N}(x_0, \sigma^2 I)$, a Gaussian around the point $x_0$ to be classified, and $\rho = \mathcal{N}(x_0 + \delta, \sigma^2 I)$, a Gaussian around $x_0$ perturbed by $\delta$. They prove that (2) holds for all classifiers $\phi$ if we define*

$$d(\mu, \rho) = \frac{1}{\sigma}\|\delta\|_2 \equiv \sqrt{2\text{KL}(\mu \parallel \rho)}, \;\; f(p) = |\Phi^{-1}(p)|, \tag{3}$$

*where $\text{KL}(\cdot)$ denotes KL divergence and $\Phi^{-1}$ denotes the inverse CDF of the Gaussian distribution.*

Although this previous work focused on the case of randomized smoothing of continuous data via Gaussian noise, this is by no means a requirement of the approach. For instance, Lee et al. (2019) considers an alternative approach for dealing with discrete variables.

**Example 2.** *The randomized smoothing guarantee of Lee et al. (2019) uses the factorized smoothing measure in $d$ dimensions $\mu_{\alpha,K}(\mathbf{x}) = \Pi_{i=1}^{d}\mu_{\alpha,K,i}(\mathbf{x}_i)$, defined with respect to parameters $\alpha \in [0, 1], K \in \mathbb{N}$, and a base input $\mathbf{z} \in \{0, \ldots, K\}^d$, where*

$$\mu_{\alpha,K,i}(\mathbf{x}_i) = \begin{cases} \alpha, & \text{if } \mathbf{x}_i = \mathbf{z}_i \\ \frac{1-\alpha}{K}, & \text{if } \mathbf{x}_i \in \{0, \ldots, K\}, \; \mathbf{x}_i \neq \mathbf{z}_i, \end{cases}$$

*with $\mathbf{x}_i$ being the $i^{th}$ dimension of $\mathbf{x}$. $\rho_{\alpha,K}$ is similarly defined for a perturbed input $\mathbf{z}'$. They guarantee that (2) holds if we define*

$$d(\mu, \rho) = r \overset{def}{=} \|\mathbf{z}' - \mathbf{z}\|_0, \;\; f(p) = \mathcal{F}_{\alpha,K,d}(\max(p, 1 - p)). \tag{4}$$

In words, the smoothing distribution is such that each dimension is independently perturbed to one of the other $K$ values uniformly at random with probability $1 - \alpha$. $\mathcal{F}$ is a combinatorial function; $\mathcal{F}_{\alpha,K,d}(p)$ is defined as the maximum number of dimensions—out of $d$ total—by which $\mu_{\alpha,K}$ and $\rho_{\alpha,K}$ can differ such that a set with measure $p$ under $\mu_{\alpha,K}$ is guaranteed to have measure at least $\frac{1}{2}$ under $\rho_{\alpha,K}$. Lee et al. (2019) prove that this value is independent of $\mathbf{z}$ and $\mathbf{z}'$.

Finally, concurrent work has considered a more general form of randomized smoothing that doesn't require strict assumptions on the distributions but is still able to provide similar guarantees.

**Example 3** (Generic bound from Anonymous (2019)). *Given any two smoothing distributions $\mu, \rho$, we have the generic randomized smoothing robustness certificate, ensuring that (2) holds with definitions*

$$d(\mu, \rho) = \text{KL}(\rho \parallel \mu), \;\; f(p) = -\frac{1}{2}\log(4p(1 - p)). \tag{5}$$

**Randomized smoothing in practice** For most classifiers, the expectation $G(\mu, \phi)$ cannot be computed exactly, and so we must resort to Monte Carlo approximation. In this "standard" form of randomized smoothing, we draw multiple random samples from $\mu$ and use these to construct a high-probability bound on $G(\mu, \phi)$ for certification. More precisely, this bound should be a *lower* bound on $G(\mu, \phi)$ when the hard prediction $g(\mu, \phi) = 1$ and an *upper* bound otherwise; this ensures in both cases that we underestimate the true certified robustness for the classifier $g$. The procedure is shown in Algorithm 2 in Appendix A. These estimates can then be plugged into a randomized smoothing robustness guarantee to provide a high probability certified robustness bound for the classifier $g$.

## 4 LABEL-FLIPPING ROBUSTNESS

We now present the main contribution of this paper, a technique for using randomized smoothing to provide certified robustness against label-flipping attacks. Specifically, we first propose a generic strategy for applying randomized smoothing to certify a prediction function against pointwise label flipping attacks. We show how this general approach can be made tractable using linear least-squares classification combined with pre-trained deep features, and we use the Chernoff inequality to analytically bound the relevant probabilities for the randomized smoothing procedure. Notably, although we are employing a randomized approach, the final algorithm does not use any random sampling, but rather relies upon a convex optimization problem to compute the certified robustness.

To motivate the approach, we note that in prior work, randomized smoothing was applied at test time with the function $\phi : \mathcal{X} \to \{0, 1\}$ being a (potentially deep) classifier that we wish to smooth.

---

**Algorithm 1** Randomized smoothing for label-flipping robustness

---

**Input:** feature mapping $h : \mathbb{R}^d \to \mathbb{R}^k$; noise parameter $q$; training set $\{(x_i, y_i) \in \mathbb{R}^d \times \{0, 1\}\}_{i=1}^n$ (with potentially adversarial labels); additional inputs to predict $\{x_j \in \mathbb{R}^d\}_{j=1}^m$
1. Pre-compute matrix $\mathbf{M}$

$$\mathbf{M} = \mathbf{X}\left(\mathbf{X}^T\mathbf{X} + \lambda\mathbf{I}\right)^{-1} \tag{6}$$

where $\mathbf{X} \equiv h(x_{1:n})$ and $\lambda = (1 + q)\frac{\hat{\sigma}^2 k}{2n}\frac{\sigma_{max}(\mathbf{X}^T\mathbf{X})}{\sigma_{min}(\mathbf{X}^T\mathbf{X})}$
**for** $j = 1, \ldots, m$ **do**
   1. Compute vector $\boldsymbol{\alpha}^j = \mathbf{M}h(x_j)^T$
   2. Compute optimal Chernoff parameter $t$ via Newton's method

$$t^\star = \arg\min_t \left\{ t/2 + \sum_{i:y_i=1} \log(q + (1-q)e^{-t\boldsymbol{\alpha}_i^j}) + \sum_{i:y_i=0} \log((1-q) + qe^{-t\boldsymbol{\alpha}_i^j}) \right\} \tag{7}$$

and let $p^\star = \max(1 - B_{|t^\star|}, 1/2)$ where $B_{|t^\star|}$ is the Chernoff bound (13) evaluated at $|t^\star|$.
**Output:** Prediction $\hat{y}_j = \mathbf{1}\{t^\star \geq 0\}$ and certification that prediction will remain constant for up to $r$ training label flips, where

$$r = \left\lfloor \frac{\log(4p^\star(1-p^\star))}{2(1-2q)\log\left(\frac{q}{1-q}\right)} \right\rfloor \tag{8}$$

(or a larger number of flips using the exact method of Lee et al. (2019)).
**end for**

---

However, there is no requirement that the function $\phi$ be a classifier at all; the theory holds for any binary-valued function. Instead of treating $\phi$ as a trained classifier, we consider $\phi$ to be *an arbitrary learning algorithm* which takes as input a training dataset $\{x_i, y_i\}_{i=1}^n \in (\mathcal{X} \times \{0, 1\})^n$ and an additional example [2] $x_{n+1}$ without a corresponding label, which we aim to predict. In other words, the combined goal of $\phi$ is to first train a classifier and then predict the label of the new example. Thus, we consider test time outputs to be a function of both the test time input and the training data that produced the classifier. This perspective allows us to reason about how changes to training data affect the classifier at test time, reminiscent of work on influence functions of deep neural networks (Koh & Liang, 2017; Yeh et al., 2018). When applying randomized smoothing in this setting, we randomize over the labels in the training set, rather than over the test-time input to be classified. Analogous to previous applications of randomized smoothing, if the majority vote of the classifiers trained with these randomly sampled labels has a large margin, it will confer a degree of adversarial robustness to some number of adversarially corrupted labels.

To formalize this intuition, consider two different assignments of $n$ training labels $Y_1, Y_2 \in \{0, 1\}^n$ which differ on precisely $r$ labels. Let $\mu$ (resp. $\rho$) be the distribution resulting from independently flipping each of the labels in $Y_1$ (resp. $Y_2$) with probability $q$. It is clear that as $r$ increases, $d(\mu, \rho)$ should also increase. In fact, it is simple to show (see Appendix B.3 for derivation) that the exact KL divergence between these two distributions is

$$\mathrm{KL}(\mu \parallel \rho) = \mathrm{KL}(\rho \parallel \mu) = r(1-2q)\log\left(\frac{1-q}{q}\right). \tag{9}$$

Plugging in the robustness guarantee (5), we have that $g(\mu, \phi) = g(\rho, \phi)$ so long as

$$r \leq \frac{\log(4p(1-p))}{2(1-2q)\log\left(\frac{q}{1-q}\right)}, \tag{10}$$

This implies that for any test point, as long as (10) is satisfied, $g$'s prediction (the majority vote weighted by the smoothing distribution) will not change if an adversary corrupts the training set

---

[2]Note that our algorithm does not actually require access to the test data to do the necessary precomputation. We present it here as such merely to give an intuitive idea of the procedure.

from $Y_1$ to $Y_2$, or indeed to any other training set that differs on at most $r$ labels. We can tune the noise hyperparameter $q$ to achieve the largest possible upper bound in (10); more noise will likely decrease the margin of the majority vote $p$, but will also decrease the divergence.

**Computing a tight bound** This approach has a simple closed form, but the bound is not tight. We can derive a tight bound via a combinatorial approach as in Lee et al. (2019). By precomputing the quantities $\mathcal{F}_{1-q,1,n}^{-1}(r)$ from Equation (4) for each $r$, we can simply compare $G(\mu, \phi)$ to each of these and thereby certify robustness to the highest possible number of label flips. This computation can be expensive, but it provides a significantly tighter robustness guarantee, certifying approximately twice as many label flips for a given bound on $G(\mu, \phi)$ (See Figure 5 in Appendix D). We make use of this tighter bound in our experiments, but we emphasize that meaningful results can be achieved even with the looser bound, which is orders of magnitude cheaper to compute.

## 4.1 Efficient implementation via least squares classifiers

There may appear to be one major impracticality of the algorithm proposed in the previous section, if considered naively: treating the function $\phi$ as an entire training-plus-single-prediction process would require that we train multiple classifiers, over multiple random draws of the labels $y$, all to make a prediction on a single example. In this section, we describe a sequence of tools we employ to restrict the architecture and training process in a manner that drastically reduces this cost, bringing it in line with the cost of classifying a single example. The full procedure, with all the parts described below, can be found in Algorithm 1.

**Linear least-squares classification** The fundamental simplifying assumption we make in this work is to restrict the "training" process done by the classifier $\phi$ to be done via a linear least-squares solve. Given the training set $\{x_i, y_i\}_{i=1}^n$, we assume that there exists some feature mapping $h : \mathbb{R}^d \to \mathbb{R}^k$ (where $k < n$), which typically would consist of a deep network pre-trained on a similar task, or possibly trained in an unsupervised fashion on $x_{1:n}$ (i.e. independent of the training labels, which are presumed to be potentially poisoned). This may seem to be a strong assumption, but similar assumptions are commonly made when using pre-trained models or in the meta-learning setting, and the transferability of pre-trained features is well documented (Donahue et al., 2014; Bo et al., 2010; Yosinski et al., 2014). Given this feature mapping, let $\boldsymbol{X} = h(x_{1:n}) \in \mathbb{R}^{n \times k}$ be the training point features and let $\boldsymbol{y} = y_{1:n} \in \{0,1\}^n$ be the labels. Our training process consists of finding the least-squares fit to the training data, i.e., we find parameters $\hat{\boldsymbol{\beta}} \in \mathbb{R}^k$ via the normal equation $\hat{\boldsymbol{\beta}} = \left(\boldsymbol{X}^T \boldsymbol{X}\right)^{-1} \boldsymbol{X}^T \boldsymbol{y}$ and then we make a prediction on the new example via the linear function $h(x_{n+1})\hat{\boldsymbol{\beta}}$. Although it may seem odd to fit a classification task with least-squares loss, binary classification with linear regression is known to be equivalent to Fisher's linear discriminant (Mika, 2003) and often works quite well in practice.

The real advantage of the least-squares approach in this setting is that it reduces the prediction to a linear function of $\boldsymbol{y}$, and thus randomizing over the labels is straightforward. Specifically, letting

$$\boldsymbol{\alpha} = \boldsymbol{X} \left(\boldsymbol{X}^T \boldsymbol{X}\right)^{-1} h(x_{n+1})^T, \tag{11}$$

the prediction $h(x_{n+1})\hat{\boldsymbol{\beta}}$ can be equivalently given by $\boldsymbol{\alpha}^T \boldsymbol{y}$ (this is effectively just the kernel representation of the linear classifier). Thus, we can simply compute $\boldsymbol{\alpha}$ one time and then randomly sample many different sets of labels in order to build a standard randomized smoothing bound. Further, we can pre-compute just the $\boldsymbol{X} \left(\boldsymbol{X}^T \boldsymbol{X}\right)^{-1}$ term and reuse it for each test point.

$\ell_2$ **regularization for better conditioning** Unfortunately, it is unlikely to be the case that the training points are well-behaved for linear regression in the feature space. To address this, we instead solve an $\ell_2$ regularized version of least-squares. This is a common tool for solving systems with ill-conditioned or random design matrices (Hsu et al., 2014; Suggala et al., 2018). Luckily, there still exists a pre-computable closed-form solution to this problem, whereby we solve

$$\boldsymbol{\alpha} = \boldsymbol{X}(\boldsymbol{X}^T \boldsymbol{X} + \lambda \boldsymbol{I})^{-1} h(x_{n+1})^T. \tag{12}$$

The other parts of our algorithm remain unchanged. Following results in Suggala et al. (2018), we set the regularization parameter $\lambda = (1+q)\frac{\hat{\sigma}^2 k}{2n}\kappa(\boldsymbol{X}^T \boldsymbol{X})$ where $\hat{\sigma}^2 = \frac{\|\boldsymbol{y} - \boldsymbol{X}\hat{\boldsymbol{\beta}}_{OLS}\|_2^2}{n-k}$ is an estimate

of the variance (Dicker, 2014) and $\kappa(\cdot)$ is the condition number equal to the ratio of the largest and smallest singular values. The $(1 + q)$ term is to help account for the variance caused by label flips.

**Efficient tail bounds via the Chernoff inequality**    Even more compelling, due to the linear structure of this prediction, we can forego a sampling-based approach entirely and directly bound the tail probabilities using Chernoff bounds. Because the underlying binary prediction function $\phi$ will output the label 1 for the test point whenever $\boldsymbol{\alpha}^T\boldsymbol{y} \geq 1/2$ and 0 otherwise, we can derive an analytical upper bound on the probability that $g$ predicts one label or the other via the Chernoff bound. By upper bounding the probability of the *opposite* prediction, we simultaneously derive a lower bound on $p$ which can be plugged in to (10) to determine the classifier's robustness. Concretely, we can upper bound the probability that the classifier outputs the label 0 by

$$P(\boldsymbol{\alpha}^T\boldsymbol{y} \leq 1/2) \leq \min_{t>0} \left\{ e^{t/2} \prod_{i=1}^{n} E[e^{-t\boldsymbol{\alpha}_i y_i}] \right\} = \min_{t>0} \left\{ e^{t/2} \prod_{i=1}^{n} q e^{-t\boldsymbol{\alpha}_i(1-y_i)} + (1-q)e^{-t\boldsymbol{\alpha}_i y_i} \right\}.$$
(13)

Conversely, the probability that the classifier outputs the label 1 has the analogous upper bound which is the same as (13) but evaluated at $-t$. Thus, we can solve the minimization problem unconstrained over $t$, and then let the sign of $t$ dictate which label to predict and the value of $t$ determine the bound. The objective (13) is log-convex in $t$ and can be easily solved by Newton's method. Note that in some cases, neither Chernoff upper bound will be less than $\frac{1}{2}$, meaning we cannot determine the true value of $g(\mu, \phi)$. In these cases, we simply define the classifier's prediction to be determined by the sign of $t$. While we can't guarantee that this classification will match the true majority vote, our algorithm will certify a robustness to 0 flips, so the guarantee is still valid. We avoid abstaining so as to assess our classifier's non-robust accuracy.

The key property we emphasize is that, unlike previous randomized smoothing applications, the final algorithm involves *no sampling whatsoever*. Instead, the prediction probabilities are bounded directly via the Chernoff bound, without any need for Monte Carlo approximation. Thus, the method is able to generate *truly certifiable* robust predictions using approximately the same complexity as traditional predictions with a deep network.

## 5    EXPERIMENTS

Following Koh & Liang (2017) and Steinhardt et al. (2017), we perform experiments on MNIST 1/7, the IMDB review sentiment dataset (Maas et al., 2011), and the Dogfish binary classification challenge taken from ImageNet. We run additional experiments on the full MNIST dataset; to the best of our knowledge, this is the first multi-class classification algorithm with certified robustness to label-flipping attacks. For each dataset and each noise level $q$ we report the *certified test set accuracy* at $r$ training label flips. That is, for each possible number of flips $r$, we plot the fraction of the test set that was both correctly classified and certified to not change under at least $r$ flips.

Because the above are binary classification tasks, one could technically achieve a certified accuracy of 50% at $r = \infty$ (or 10% for MNIST) by letting $g$ be constant. A constant classifier would be infinitely robust, but it is not a very meaningful baseline. However, we include the accuracy of such a classifier in our plots (black dotted line) as a reference.

To properly justify the need for such certified defenses, and to get a sense of the scale of our certifications, we also generated label-flipping attacks against the undefended binary MNIST and Dogfish models. Following previous work, the undefended models were implemented as convolutional neural networks, trained on the clean data, with all but the top layer frozen—this is equivalent to logistic regression on the learned features. For each test point we recorded how many flips were required to change the network's prediction. This number serves as an upper bound for the robustness of the network on that test point, but we note that our attacks were quite rudimentary and could almost certainly be improved upon to tighten this upper bound. Appendix C contains the details of our attack implementations.

**Results on MNIST**    The MNIST 1/7 dataset (LeCun et al., 1998) consists of just the classes 1 and 7, totalling 13007 training points and 2163 test points. We trained a simple convolutional neural network on the other eight MNIST digits to learn a 50-dimensional feature embedding and then

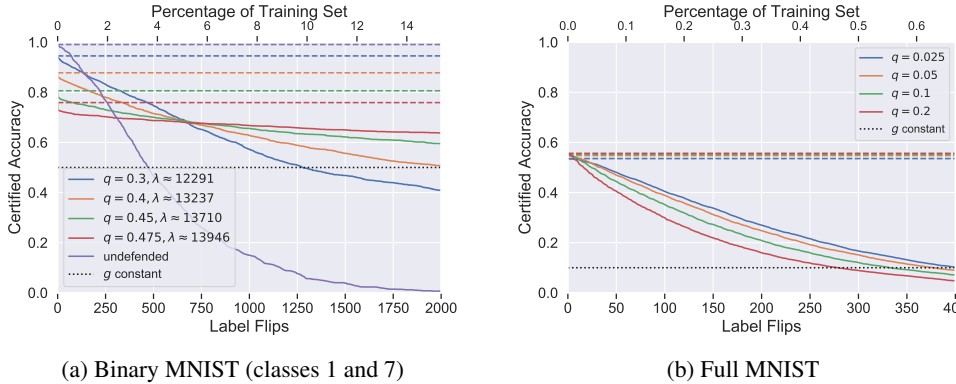

(a) Binary MNIST (classes 1 and 7)  (b) Full MNIST

Figure 1: MNIST 1/7 ($n = 13007$, left) and full MNIST ($n = 60000$, right) test set certified accuracy to adversarial training label flips as $q$ is varied. The hyperparameter $q$ controls a robust/non-robust accuracy trade-off. The solid lines represent certified accuracy, except for the undefended classifier which represents our attack. The dashed lines of the same color are the overall non-robust accuracy of each classifier. The black dotted line is the performance of a constant classifier, assuming equal representation of the classes.

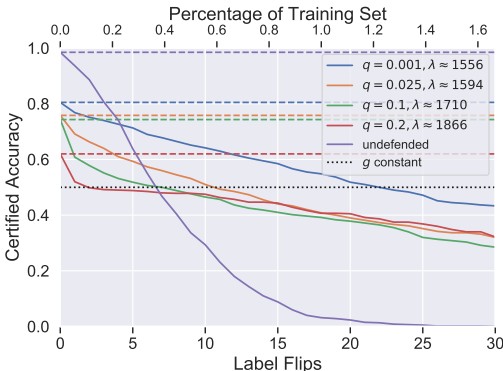

Figure 2: Dogfish ($n = 1800$) test set certified accuracy to adversarial label flips as $q$ is varied. The classifiers' certified accuracy curves cross each other further along the x-axis.

calculated Chernoff bounds for $G(\mu, \phi)$ as described in Section 4.1. Figure 1a displays the certified accuracy on the test set for varying probabilities $q$. As in prior work on randomized smoothing, the noise parameter $q$ balances a trade-off; as $q$ increases, the required margin $|G(\mu, \phi) - \frac{1}{2}|$ to certify a given number of flips decreases. On the other hand, this results in more noisy training labels, which reduces the margin and therefore results in lower robustness and often lower accuracy. Figure 1b depicts the certified accuracy for the full MNIST test set. See Appendix B for derivations of the bounds and optimization algorithm in the multi-class case. In addition to this being a significantly more difficult classification task, our pre-trained classifier could not rely on features learned from other handwritten digits; instead, we used the features from a network trained on the Omniglot dataset (Lake et al., 2015), as the task of classifying handwritten characters is quite similar. Despite the lack of fine-tuned features, our algorithm still achieves significant certified accuracy under a large number of adversarial label flips. We observed that regularization did not make a large difference for the multi-class case, possibly due to the inaccuracy of the residual term when calculating the noise estimate.

See Figure 4 in Appendix D for the effect of $\ell_2$ regularization on the OLS solution. We note that at a slight cost to non-robust accuracy, the regularization results in substantially higher certified accuracy at almost all radii. A similar effect was observed for all our datasets and at all values of $q$.

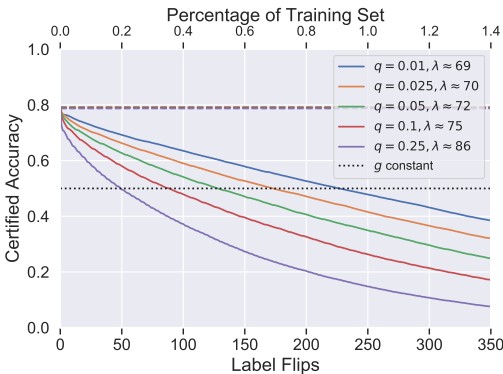

Figure 3: IMDB Review Sentiment ($n = 25000$) test set certified accuracy. The non-robust accuracy slightly decreases as $q$ increases; for $q = 0.01$ the non-robust accuracy is 79.108%, while for $q = 0.25$ it is 78.732%.

**Results on Dogfish**   The Dogfish dataset contains images from the ImageNet dog and fish synsets, 900 training points and 300 test points from each. We trained a ResNet-50 (He et al., 2016) on the standard ImageNet training set but removed all images labeled as any kind of dog or fish. Our pre-trained feature embedder was therefore capable of extracting image features but had no concept of features specific to either class. We used PCA to reduce the 2048 dimensional feature space to 1700 dimensions before solving, to ensure the system was not underdetermined. Figure 2 displays the results of our poisoning attack along with our certified defense. Under the undefended model, more than 50% of the test points can be successfully attacked with no more than 7 label flips, whereas our model with $q = 0.001$ can certifiably correctly classify 67.83% of the test points under the same attack. It would take more than three times as many flips for *each test point individually* to push our model to less than 50% certified accuracy, at which point the undefended classifier would be reduced to at most 0.83%.

Because the predictions on the test points can be changed by flipping such a small fraction of the training set, high values of $q$ reduce the randomized models' predictions to almost pure chance—this means we are unable to achieve the margins necessary to certify a large number of flips. We therefore found that smaller levels of noise were necessary to achieve high certified test accuracy. This suggests that the more susceptible the original, non-robust classifier is to label flips, the lower $q$ should be set for the corresponding randomized classifier.

**Results on IMDB**   Figure 3 plots the result of our randomized smoothing procedure on the IMDB review sentiment dataset. This dataset contains 25,000 training examples and 25,000 test examples, evenly split between "positive" and "negative". To extract the features we applied the Google News pre-trained Word2Vec to all the words in each review and averaged them. This feature embedding is considerably noisier than that of an image dataset, as most of the words in a review are irrelevant to sentiment classification. Indeed, Steinhardt et al. (2017) also found that the IMDB dataset was much more susceptible to adversarial corruption than images when using bag-of-words features. Consistent with this observation, we found smaller levels of noise to result in larger certified accuracy. We expect significant improvements could be made with a more refined choice of feature embedding.

We also observed that increasing $q$ does not significantly decrease the non-robust accuracy. We believe this is because the limitation of these classifiers is not the label noise, but rather the difficulty in classifying the imperfect feature embeddings with linear regression. Thus, the sign of $t^\star$ was frequently correct and remained on the same side of 0 as $q$ increased, but the margin of $G(\mu, \phi)$ decreased too rapidly to certify a large number of flips. This was similarly observed in the case of full MNIST, where the pre-trained features came from a network trained on the Omniglot dataset and therefore were not as informative for classification.

## 6 CONCLUSION

In this work we have presented a certified defense against a strong class of adversarial label-flipping attacks where an adversary can flip labels to cause a misclassification on each test point separately. This contrasts with previous data poisoning settings, which have typically only considered an adversary who wishes to degrade the classifier's accuracy on the test distribution as a whole, and it brings the adversary's objective more in line with that of backdoor attacks and test-time adversarial perturbations. Leveraging randomized smoothing, a method originally developed for certifying robustness to test-time perturbations, we presented a classifier that can be certified robust to these pointwise train-time attacks. We then offered a tractable algorithm for evaluating this classifier which, despite being rooted in randomization, can be computed with no Monte Carlo sampling whatsoever, resulting in a truly certifiably robust classifier. This algorithm results in the first multi-class classification algorithm that is certifiably robust to label-flipping attacks.

There are several avenues for improvement to this line of work, perhaps the most immediate being the method for learning the input features. For example, rather than considering fixed features generated by a pre-trained deep network, extensions could leverage neural tangent kernels (Jacot et al., 2018) to also allow for efficient learning with perturbed inputs or more flexible representations. Using this or other approaches, the analysis could be extended to other types of smoothing distributions applied to the training data, such as randomizing over the input features to provide robustness to more general data poisoning attacks. Conversely, we also hope that our defense to this threat model will inspire the development of more powerful train-time attacks, against which future defenses can be evaluated.

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

## A    GENERIC RANDOMIZED SMOOTHING ALGORITHM

---

**Algorithm 2** Generic randomized smoothing procedure

---

**Input:** function $\phi : \mathcal{X} \to \{0, 1\}$, number of samples $N$, smoothing distribution $\mu$, failure probability $\delta > 0$

**for** $i = 1, \ldots, N$ **do**
   Sample $x_i \sim \mu$ and compute $y_i = \phi(x_i)$
**end for**
Compute approximate smoothed output

$$\hat{g}(\mu, \phi) = \mathbf{1}\left\{ \frac{1}{N} \sum_{i=1}^{N} y_i \geq 1/2 \right\} \tag{14}$$

Compute bound $\hat{G}(\mu, \phi)$ such that with probability $1 - \delta$

$$\hat{G}(\mu, \phi) \begin{cases} \leq G(\mu, \phi) & \text{if } \hat{g}(\mu, \phi) = 1 \\ \geq G(\mu, \phi) & \text{if } \hat{g}(\mu, \phi) = 0 \end{cases} \tag{15}$$

**Output:** Prediction $\hat{g}(\mu, \phi)$ and probability bound $\hat{G}(\mu, \phi)$, or abstention if $\hat{g}(\mu, \phi)$ and $\hat{G}(\mu, \phi)$ are on different sides of $\frac{1}{2}$.

---

## B    THE MULTI-CLASS SETTING

Although the notation and algorithms are slightly more complex, all the methods we have discussed in the main paper can be extended to the multi-class setting. In this case, we consider a class label $y \in \{1, \ldots, K\}$, and we again seek some smoothed prediction such that the classifier's prediction on a new point will not change with some number $r$ flips of the labels in the training set.

### B.1    RANDOMIZED SMOOTHING IN THE MULTI-CLASS CASE

We here extend our notation to the case of more than two classes. Recall our original definition of $G$,

$$G(\mu, \phi) = \mathbf{E}_{x \sim \mu}[\phi(x)] = \int_{\mathcal{X}} \mu(x)\phi(x)dx,$$

where $\phi : \mathcal{X} \to \{0, 1\}$. More generally, consider a classifier $\phi : \mathcal{X} \to [K]$, outputting the index of one of $K$ classes. Under this formulation, for a given class $c \in [K]$, we have

$$G(\mu, \phi, c) = \mathbf{E}_{x \sim \mu}[\phi_c(x)] = \int_{\mathcal{X}} \mu(x)\phi_c(x)dx, \tag{16}$$

where $\phi_c(x) = \mathbf{1}\{\phi(x) = c\}$ is the indicator function for if $\phi(x)$ outputs the class $c$. In this case, the hard threshold $g$ is evaluated by returning the class with the highest probability. That is,

$$g(\mu, \phi) = \arg\max_c G(\mu, \phi, c). \tag{17}$$

## B.2 LINEARIZATION AND CHERNOFF BOUND APPROACH FOR THE MULTICLASS CASE

Using the same linearization approach as in the binary case, we can formulate an analogous approach which forgoes the need to actually perform random sampling at all and instead directly bounds the randomized classifier using the Chernoff bound.

Adopting the same notation as in the main text, the equivalent least-squares classifier for the multi-class setting finds some set of weights

$$\hat{\boldsymbol{\beta}} = \left(\boldsymbol{X}^T \boldsymbol{X}\right)^{-1} \boldsymbol{X}^T \boldsymbol{Y} \tag{18}$$

where $\boldsymbol{Y} \in \{0,1\}^{n \times K}$ is a binary matrix with each row equal to a one-hot encoding of the class label (note that the resulting $\hat{\boldsymbol{\beta}} \in \mathbb{R}^{k \times K}$ is now a matrix, and we let $\hat{\boldsymbol{\beta}}_i$ refer to the $i$th column). At prediction time, the predicted class of some new point $x_{n+1}$ is simply given by the prediction with the highest value, i.e.,

$$\hat{y}_{n+1} = \arg\max_i \hat{\boldsymbol{\beta}}_i^T h(x_{n+1}). \tag{19}$$

Alternatively, following the same logic as in the binary case, this same prediction can be written in terms of the $\boldsymbol{\alpha}$ variable as

$$\hat{y}_{n+1} = \arg\max_i \boldsymbol{\alpha}^T \boldsymbol{Y}_i \tag{20}$$

where $\boldsymbol{Y}_i$ denotes the $i$th column of $\boldsymbol{Y}_i$.

In our randomized smoothing setting, we again propose to flip the class of any label with probability $q$, selecting an alternative label uniformly at random from the remaining $K - 1$ labels. Assuming that the predicted class label is $i$, we wish to bound the probability that

$$P(\boldsymbol{\alpha}^T \boldsymbol{Y}_i < \boldsymbol{\alpha}^T \boldsymbol{Y}_{i'}) \tag{21}$$

for all alternative classes $i'$. By the Chernoff bound, we have that

$$\log P(\boldsymbol{\alpha}^T \boldsymbol{Y}_i < \boldsymbol{\alpha}^T \boldsymbol{Y}_{i'}) = \log P(\boldsymbol{\alpha}^T (\boldsymbol{Y}_i - \boldsymbol{Y}_{i'}) \le 0)$$

$$\le \min_{t>0} \left\{ \sum_{j=1}^n \log \mathbf{E} \left[ e^{-t\boldsymbol{\alpha}_j (\boldsymbol{Y}_{ji} - \boldsymbol{Y}_{ji'})} \right] \right\}. \tag{22}$$

The random variable $\boldsymbol{Y}_{ji} - \boldsymbol{Y}_{ji'}$ takes on three different distributions depending on if $y_j = i$, if $y_j = i'$, or if $y_j \ne i$ and $y_j \ne i'$. Specifically, this variable can take on the terms $+1, 0, -1$ with the associated probabilities

$$P(\boldsymbol{Y}_{ji} - \boldsymbol{Y}_{ji'} = +1) = \begin{cases} 1-q & \text{if } y_j = i, \\ q/(K-1) & \text{otherwise.} \end{cases}$$

$$P(\boldsymbol{Y}_{ji} - \boldsymbol{Y}_{ji'} = -1) = \begin{cases} 1-q & \text{if } y_j = i', \\ q/(K-1) & \text{otherwise.} \end{cases} \tag{23}$$

$$P(\boldsymbol{Y}_{ji} - \boldsymbol{Y}_{ji'} = 0) = \begin{cases} q(K-2)/(K-1) & \text{if } y_j = i \text{ or } y_j = i', \\ 1 - 2q/(K-1) & \text{otherwise.} \end{cases}$$

Combining these cases directly into the Chernoff bound gives

$$\log P(\boldsymbol{\alpha}^T \boldsymbol{Y}_i < \boldsymbol{\alpha}^T \boldsymbol{Y}_{i'}) \le \min_{t>0} \Bigg\{ \sum_{j:y_j=i} \log \left( (1-q)e^{-t\boldsymbol{\alpha}_j} + q\frac{K-2}{K-1} + \frac{q}{K-1} e^{t\boldsymbol{\alpha}_j} \right) +$$

$$\sum_{j:y_j=i'} \log \left( \frac{q}{K-1} e^{-t\boldsymbol{\alpha}_j} + q\frac{K-2}{K-1} + (1-q)e^{t\boldsymbol{\alpha}_j} \right) +$$

$$\sum_{j:y_j \ne i, y_j \ne i'} \log \left( \frac{q}{K-1} e^{-t\boldsymbol{\alpha}_j} + 1 - 2\frac{q}{K-1} + \frac{q}{K-1} e^{t\boldsymbol{\alpha}_j} \right) \Bigg\}. \tag{24}$$

Again, this problem is convex in $t$, and so can be solved efficiently using Newton's method. And again since the reverse case can be computed via the same expression we can similarly optimize this in an unconstrained fashion. Specifically, we can do this for every pair of classes $i$ and $i'$, and return the $i$ which gives the smallest lower bound for the worst-case choice of $i'$.

### B.3 KL DIVERGENCE BOUND

To compute actual certification radii, we will derive the KL divergence bound for the the case of $K$ classes. Let $\mu, \rho$ be defined as in Section 4, except that as in the previous secftion when a label is flipped with probability $q$ it is changed to one of the other $K - 1$ classes uniformly at random. Let $\mu_i$ and $\rho_i$ refer to the independent measures on each dimension which collectively make up the factorized distributions $\mu$ and $\rho$ (i.e., $\mu(x) = \prod_{i=1}^{d} \mu_i(x)$). Further, let $Y_1^i$ be the $i^{th}$ element of $Y_1$, meaning it is the "original" class which may or may not be flipped when sampling from $\mu$. First noting that each dimension of the distributions $\mu$ and $\rho$ are independent, we have

$$
\begin{aligned}
\mathrm{KL}(\rho \parallel \mu) &= \sum_{i=1}^{n} \mathrm{KL}(\rho_i \parallel \mu_i) \\
&= \sum_{i:\rho_i \neq \mu_i} \mathrm{KL}(\rho_i \parallel \mu_i) \\
&= r \left( \sum_{j=1}^{K} \rho_i(j) \log \left( \frac{\rho_i(j)}{\mu_i(j)} \right) \right) \\
&= r \left( \rho_i(Y_1^i) \log \left( \frac{\rho_i(Y_1^i)}{\mu_i(Y_1^i)} \right) + \rho_i(Y_2^i) \log \left( \frac{\rho_i(Y_2^i)}{\mu_i(Y_2^i)} \right) \right) \\
&= r \left( (1-q) \log \left( \frac{1-q}{\frac{q}{K-1}} \right) + \frac{q}{K-1} \log \left( \frac{\frac{q}{K-1}}{1-q} \right) \right) \\
&= r \left( 1 - \frac{Kq}{K-1} \right) \log \left( \frac{(1-q)(K-1)}{q} \right)
\end{aligned}
$$

Plugging in the robustness guarantee (5), we have that $g(\mu, \phi) = g(\rho, \phi)$ so long as

$$
r \leq \frac{\log(4p(1-p))}{2(1 - \frac{Kq}{K-1}) \log \left( \frac{q}{(1-q)(K-1)} \right)}. \tag{25}
$$

Setting $K = 2$ recovers the divergence term (9) and the bound (10).

## C  DESCRIPTION OF LABEL-FLIPPING ATTACKS ON MNIST 1/7 AND DOGFISH

Due to the dearth of existing work on label-flipping attacks for deep networks, our attacks on MNIST and Dogfish were quite straightforward; we expect significant improvements could be made to tighten this upper bound.

For Dogfish, we used a pretrained Inception network (Szegedy et al., 2016) to evaluate the influence of each training point with respect to the loss of each test point (Koh & Liang, 2017). As in prior work, we froze all but the top layer of the network for retraining. Once we obtained the most influential points, we flipped the first one and recomputed approximate influence using only the top layer for efficiency. After each flip, we recorded which points were classified differently and maintained for each test point the successful attack which required the fewest flips. When this was finished, we also tried the reverse of each attack to see if any of them could be achieved with even fewer flips.

For MNIST we implemented two similar attacks and kept the best attack for each test point. The first attack simply ordered training labels by their $\ell_2$ distance from the test point in feature space, as a proxy for influence. We then tried flipping these one at a time until the prediction changed, and we also tried the reverse. The second attack was essentially the same as the Dogfish attack, ordering the test points by influence. To calculate influence we again assumed a frozen feature map; specifically, using the same notation as Koh & Liang (2017), the influence of flipping the label of a training point

$z = (x, y)$ to $z^- = (x, 1 - y)$ on the loss at the test point $z_{\text{test}}$ is:

$$\frac{dL(z_{\text{test}}, \hat{\theta}_{\epsilon, z^-, -z})}{d\epsilon} = \nabla_\theta L(z_{\text{test}}, \hat{\theta})^T \frac{d\hat{\theta}_{\epsilon, z^-, -z}}{d\epsilon}$$

$$\approx -\nabla_\theta L(z_{\text{test}}, \hat{\theta})^T H_{\hat{\theta}}^{-1} \left( \nabla_\theta L(z^-, \hat{\theta}) - \nabla_\theta L(z, \hat{\theta}) \right)$$

For logistic regression, these values can easily be computed in closed form.

## D   ADDITIONAL PLOTS

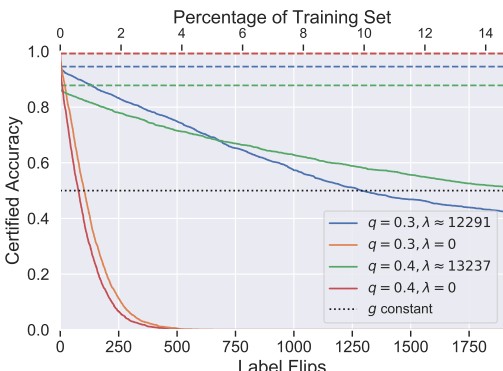

Figure 4: MNIST 1/7 test set certified accuracy with and without $\ell_2$ regularization in the computation of $\alpha$. Note that the unregularized solution achieves almost 100% non-robust accuracy, but certifies significantly lower robustness. This implies that the "training" process is not robust enough to label noise, hence the lower margin by the ensemble. In comparison, the regularized solution achieves significantly higher margins, at a slight cost in overall accuracy.

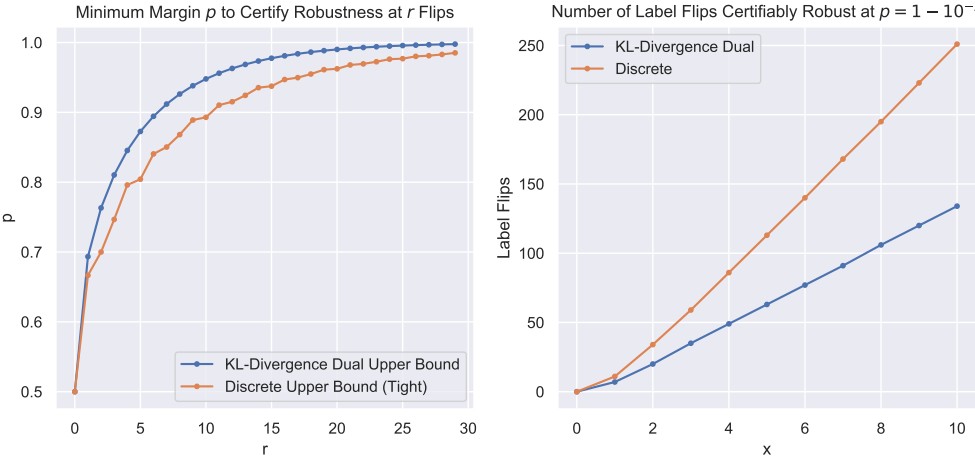

Figure 5: **Left:** Required margin $p$ to certify a given number of label flips using the generic KL bound (10) versus the tight discrete bound (4). **Right:** The same comparison, but inverted, showing the certifiable robustness for a given margin. The tight bound certifies robustness to approximately twice as many label flips.

