# OpenReview forum: "Certified Robustness to Adversarial Label-Flipping Attacks via Randomized Smoothing"
_ICLR.cc/2020/Conference — Reject_

### Official Review · AnonReviewer1 · 2019-10-22
**Official Blind Review #1**

**Rating:** 3

**Review:**

This paper leverages the randomized smoothing technique, on labels of images during training, to counter the adversarial label-flipping attack. The authors proposed a strategy to build classifiers that are certifiably robust against a strong variant of label-flipping attack that can target each test example independently. The resulting classifier can make a prediction and includes certification for each test point. On a simple binary classification problem, the proposed defensive significantly improves accuracy towards the label-flipping attack. I have the following questions about this work:

1. Can the authors provide training and testing CPU time and memory consumption, and compare them with the training without defense?

2. Can the authors generalize the proposed adversarial defense to the multi-class classification problem?

3. Can the authors also test the algorithms on the MNIST and CIFAR10 dataset with multiple different selected pairs?

4. Besides randomized smoothing on the input images, recently Wang et al showed that randomize the deep nets can
also improve the deep nets and they gave it a nice theoretical intepretation. Here is the reference: Bao Wang, Binjie Yuan, Zuoqiang Shi, Stanley J. Osher. ResNets Ensemble via the Feynman-Kac Formalism to Improve Natural and Robust Accuracies, arXiv:1811.10745, NeurIPS, 2019

Overall, this paper studies an interesting and important certified adversarial defense against label-flipping attack problems with a focus on certification on each test data, but more experimental verification is needed. Please address the above questions in rebuttal.

**Experience Assessment:**

I have published one or two papers in this area.

**Review Assessment: Checking Correctness Of Derivations And Theory:**

I carefully checked the derivations and theory.

**Review Assessment: Checking Correctness Of Experiments:**

I carefully checked the experiments.

**Review Assessment: Thoroughness In Paper Reading:**

I read the paper thoroughly.

---

> ### Author Response · Authors · 2019-11-06
> **Thanks for your comments!**
>
> Thanks for your comments! Below we respond to all of these, and hope that the clarifications can improve your evaluation of the paper.
>
> 1. The cost of training our algorithm is simply the cost of training the pre-trained classifier. In other words, there is zero increase to training time, and in fact in some cases the training time could be reduced to nothing if an appropriate pre-trained network already exists. We did not include time/memory usage for test time evaluation because it was negligible; the memory usage is no more than a standard OLS solve plus the cost of a one-dimensional second-order convex optimization problem, and the computation time is similarly negligible; a standard laptop classifies and certifies an example in ~0.01 seconds (this is just the cost of the optimization).
>
> 2. We do indeed generalize our result to the multi-class case, see the Appendix for derivations of the bounds.
>
> 3. MNIST 1/7 is the standard binary classification test set for data poisoning attacks because 1 and 7 are somewhat similar in appearance. This test set serves as a “worst case” among all two-class subsets of MNIST, since any other two classes will be significantly more distinct. We believe the ImageNet Dogfish dataset is sufficient to demonstrate our method’s applicability to more complex images, especially because ImageNet is considerably more complex than CIFAR10, and existing work similarly performs experiments using these two datasets.
>
> 4. Thank you for this reference! We will be sure to compare our work to this one in an updated version.

---

> > ### Comment · AnonReviewer1 · 2019-11-13
> > **Thank you for your reply**
> >
> > I have read your reply carefully. But I have not seen the comparison results yet.

---

> > > ### Author Response · Authors · 2019-11-14
> > > **We have just updated the paper**
> > >
> > > Please see the new version for an update to the paper which includes the multi-class case! Please let us know if there are other clarifications we can make or concerns we can address.

---

### Official Review · AnonReviewer3 · 2019-10-22
**Official Blind Review #3**

**Rating:** 3

**Review:**

Summary.

The paper investigates data poisoning type of attack. In such attacks, an adversary can alter/flip the labels of some of the training examples. The paper proposed a new approach towards certified robustness against this type of attack. In particular, the new classifier will output a prediction along with a certificate in which the prediction would not change if certain number of labels in the training data were flipped.

The authors use randomized smoothing on a binary linear classifier with logistic loss and deduce a radius of certification that is a function of the probability of flipping labels in the training data and the probabilistic separation p.


Major concerns.

1) I find the application of certification to data poisoning type adversarial attacks rather limited and is not of a major interest to the verification/certification community. Certification arose as a major issue since models implemented in practice can be fooled if subjected to noise at testing time. However, the proposed certification is on the flip rate of the labels in the training data. Since this is fixed, certification in this context is not of a major interest. The only potential relevance of such a problem is upon training models in a federated approach (online learning) even then, one can argue that the portion of the data that the adversary has access to is a small portion to the complete dataset. It feels to be that certification was somehow forced into data poisoning type of adversarial attacks although they jointly do not make much sense.



 While I do appreciate the work from authors, I am not convinced that certification in this context makes significant sense and has interest to only small group of researchers.


2) There are some serious limitation in the work specifically that the bounds are derived for a binary classifier. While the authors did indeed discuss the multi-class case, there are no experiments beyond the binary classification. The dataset sets where the sentiment analysis of IMDB, MNIST1/7 and the dog fish classes from ImageNet. Experiments on multi-class case is essential here for practical reasons.

3) May the authors clarify some few things in the experiments for me.

If I understand Figures 1,2,3 correctly, then what the authors do is that they train networks over different number of label flipping (shown as a percentage on the top of the figure). Then for each test example, they compute the maximum radius given in Eq 10 for multiple qs. If the number of flipping in the dataset is less than r (less than the maximum radius per sample) then the sample is certified.

	Question. I do not understand why is an example considered certifiable when it is both correctly classified and unaffected under at least r flips? In page 7 of the experiments section, the authors say "we plot the fraction of the test set that was both correctly classified and certified to not change under at least r flips".

	Question. Can the authors comment on the expected performance of the network upon comparing certified accuracy with networks having q as probability flip rate and when the training data flip rate percentage is exactly q. Will the certified accuracy for q that matches the flip rate in the dataset be better than the ones with q that does not match the training flip rate. For instance, in Figure 1, note that the percentage in the flip rate of the training dataset ranges from 0 to 14%, however the certified accuracy was for q that ranges from 30 to 47.5. Will the certified accuracy at a given percentage of the data be the highest for q that match the flip rate?


	Question. Since the classifier is linear (since features are fixed), can the authors comment perhaps on the relation between p and q?





Minor comments.

1) The authors need to define what p is. The authors can correct me if I'm wrong, p is a lower bound on the separation of the probability for some class, i.e.  Prop(f(x) = 1) >= p. This is similar to Cohen et al. 19. This was not defined and come as a surprise in Eq 3.

2) Moreover, in the work of Cohen et al. the smoothed classifier g is smoothed in probability which is unlike definition 1. Perhaps definition 1 fits more the framework of Lecuyer et al 19 as they showed that if an algorithm A is differentially private (probability distribution do not change much under database perturbations), the expectation over the algorithm is also differentially private.

3) In the solution to the ridge regression, \alpha is a function of the tested sample, e.g. eq 11. This can be confusing and I advise authors to consider adding a superscript \alpha^j indicating that this is a function of the test sample j. Similarly, in the algorithm \alpha in Eq 7 and bullet point 1.

4) In example 3, the paper cited is a concurrent submission to ICLR20 and not 19.

**Experience Assessment:**

I have published one or two papers in this area.

**Review Assessment: Checking Correctness Of Derivations And Theory:**

I assessed the sensibility of the derivations and theory.

**Review Assessment: Checking Correctness Of Experiments:**

I carefully checked the experiments.

**Review Assessment: Thoroughness In Paper Reading:**

I read the paper thoroughly.

---

> ### Author Response · Authors · 2019-11-06
> **Thank you for your detailed response!**
>
> Thank you for your detailed response! We will do our best to address each of your concerns. Several of your comments appear to be due to a fundamental misunderstanding of the algorithm proposed in this paper and its use; we hope that you will carefully consider our rebuttal and perhaps reevaluate the paper in light of our clarifications.
>
> 1) We respectfully disagree with your assessment of the relevance of this work to the certification community. The purpose of defense certification is to ensure that a particular algorithm/classification remains robust, even against an unknown future adversary with arbitrary compute power. With the rise of deep learning, model/algorithm certification has lagged significantly behind performance on large, complex datasets, hence the current frenzy of research into certifiable deep networks. We believe we make a significant contribution to this field. To support our claim, we note that there is a **very large body of existing work** on certified defenses to adversarial data poisoning attacks (see the related work section of our paper for many examples). Further, we believe you may have misunderstood the contribution of this paper:
>
> “However, the proposed certification is on the flip rate of the labels in the training data. Since this is fixed, certification in this context is not of a major interest.”
>
> The label flipping is a randomization tool used by our algorithm to certify robustness against *adversarial, non-fixed* label attacks. We do not certify against fixed label flips, we certify against *individual, targeted* attacks that can flip some number of labels to cause a misclassification on each test point individually.
>
> 2) We emphasize that we do not just discuss the multi-class case, we actually derive bounds for it in precisely the same manner as for the binary case in the Appendix. Thus the theoretical contribution remains applicable to the general case. We agree the paper could be strengthened with results for the multi-class case, but because a great deal of past work in this setting has focused on the binary case, we highlighted that setting here. However, we can certainly run the multi-class setting that we describe in the appendix. We will update the paper with results, ideally by the end of the ICLR review period.
>
> 3) We believe your understanding of the figures and the general algorithm is incorrect, and this appears to be the source of the majority of your concerns with this work.
>
> “...what the authors do is that they train networks over different number of label flipping (shown as a percentage on the top of the figure)”
>
> This interpretation is incorrect. The top axis is the same as the bottom axis; we plot the fraction of the test set which our algorithm certifiably correctly classifies (the y-axis) against an adversary who is allowed to adversarially flip some number of training points X (the x-axis). On the bottom, we express X as a number, and on the top we express that same number X as a percentage of the training set size, e.g., for MNIST 1/7, an adversary who can flip 1000 labels / approximately 8% of the training set can reduce our classifier with q = 0.475 to no less than ~70% accuracy.
>
> Additionally, we do not train networks over different numbers of label flips. A key aspect of our algorithm is that only one network is required; the label flipping comes into play only in the calculation of the Chernoff bounds for certification.
>
> “I do not understand why is an example considered certifiable when it is both correctly classified and unaffected under at least r flips?”
>
> This is a standard metric for certifiable classification. It indicates that under a threat model which allows the adversary to flip up to some number of labels, our classifier will certifiably achieve a particular accuracy. See [1, 2, 3, 4].
>
> “Can the authors comment on the expected performance of the network upon comparing certified accuracy with networks having q as probability flip rate and when the training data flip rate percentage is exactly q.”
>
> As above, q is not the adversarial flip rate, q is a *hyperparameter of our algorithm* which we use to randomly flip labels in the training set. The only way the flip rate percentage could be exactly q is by pure chance, because our algorithm requires that each label is flipped independently with probability q.
>
> “Since the classifier is linear (since features are fixed), can the authors comment perhaps on the relation between p and q?”
>
> We address this at several points in our paper. As q increases, the margin p decreases, but the margin necessary to certify a particular number of flips goes down as well, hence the trade-off. The exact nature of this inverse relationship varies heavily with which labels specifically are flipped (which is done randomly at test time).

---

> > ### Author Response · Authors · 2019-11-06
> > **Followup response to address minor concerns**
> >
> > 4)	To address your minor comments:
> >
> > (1) In equation (3), p is simply the input to the function. It’s like “x” in the expression “f(x)”. We note that we give the input to the function in equation (2) (i.e., G(µ, φ), which is itself defined in equation (1)).
> >
> > (2) The classifier in equation (1) is smoothed in probability. In the notation of Cohen et. al, G(µ, φ) is precisely p and g(µ, φ) is precisely the output of the smoothed classifier. Our formulation is a special case where we consider the probability that a binary classifier outputs a 0 or 1---note that the probability of a binary variable being equal to 1 is exactly its expectation.
> >
> > (3) Thank you for the suggestion, we will include this in our updated version.
> >
> > (4) Our bibliography lists this work as submitted to ICLR 20, however the paper was released in 2019.
> >
> >
> > [1]	Mathias Lecuyer, Vaggelis Atlidakis, Roxana Geambasu, Daniel Hsu, and Suman Jana. Certified robustness to adversarial examples with differential privacy.
> > [2]	Bai Li, Changyou Chen, Wenlin Wang, and Lawrence Carin. Certified adversarial robustness with additive gaussian noise.
> > [3] 	Jeremy Cohen, Elan Rosenfeld, and J. Zico Kolter. Certified adversarial robustness via randomized smoothing.
> > [4] 	Hadi Salman, Greg Yang, Jerry Li, Pengchuan Zhang, Huan Zhang, Ilya Razenshteyn, and Sebastien Bubeck. Provably robust deep learning via adversarially trained smoothed classifiers.

---

> ### Author Response · Authors · 2019-11-14
> **We have updated the paper to include multi-class experiments**
>
> The multi-class experiments have been added to the revision. In light of this and our clarifications regarding the paper, we hope you will reconsider your evaluation. Are there any other concerns or points of clarification you would like us to address?

---

### Official Review · AnonReviewer2 · 2019-10-23
**Official Blind Review #2**

**Rating:** 3

**Review:**

Summary:

This paper proposes a certifiable defense against data poisoning attacks by using a randomized smoothing approach. An adversary in such a setting is permitted to flip any r labels from a dataset of size n. The smoothing procedure (stated roughly) is to train on a dataset with "noisy" or "smoothed" labels, obtained by flipping each label with some probability q. The authors obtain a lower bound on r in terms of q. Directly using this technique requires training multiple classifiers on multiple noisy datasets. To show that this method is useful, the authors study the effectiveness of this model against a classifier that performs linear regression on a pre-trained feature extractor.

The authors provide a succinct summary of current research concerning randomized smoothing. The novelty in this paper is that it considers randomized smoothing defenses for data poisoning (label flipping) attacks, as opposed to perturbation based attacks. While the paper was an enjoyable read, I recommend rejecting the paper due to the following shortcomings that:

(1) The paper is essentially studying (a variant of) linear regression. This was really not obvious from the title or the abstract. It meant that as a reader, I had high expectations but was let down upon reading section 4.1.

(2) What prior work exists for data poisoning attacks against linear regression or logistic regression? How does the contribution in this paper fit in that context?  Adding some discussion along these lines to the paper seems necessary.

(3) There is no explanation why the authors chose the specific datasets that they studied.

**Experience Assessment:**

I have read many papers in this area.

**Review Assessment: Checking Correctness Of Derivations And Theory:**

I assessed the sensibility of the derivations and theory.

**Review Assessment: Checking Correctness Of Experiments:**

I assessed the sensibility of the experiments.

**Review Assessment: Thoroughness In Paper Reading:**

I made a quick assessment of this paper.

---

> ### Author Response · Authors · 2019-11-06
> **Thank you for your comments!**
>
> Thank you for your comments! We hope our response will suffice for a reevaluation of our paper.
>
> (1) We apologize for the misunderstanding regarding the contributions of our paper. We were aware of the possible misinterpretation, and so we tried to make it clear as early as possible what exactly we are proposing. We first mention it in the third paragraph of the introduction, but we will update the abstract to make it even more clear. However, even though we’re using the solution to ordinary least squares, we’re actually doing least-squares classification. Further, we note that this work remains the first meaningful certifiable defense applicable to deep networks. This is because unlike existing work, it leverages the approximate linear separability of the data in the feature space, but the meaningfulness of its certifications does not rely upon it.
>
> (2) We point the reviewer to the Certified Defenses paragraph of our related work section for a discussion on existing defenses for linear/logistic regression. As we noted above, these prior works can be contrasted to ours in that the usefulness of their certifications relies upon the linearity of the data (i.e., if the features cannot be fit well by a linear model, their certifications are meaningless). Further, these works provide general robustness to attacks which attempt to degrade performance on the entire test set; we provide certified robustness to targeted attacks, where the adversary can produce a separate label-flipping attack for each test point. To our knowledge, ours is the first such certified defense.
>
> (3) The datasets we chose are established test sets for literature on data poisoning attacks. See [1, 2, 3, 4, 5]
>
> [1] 	Pang Wei Koh and Percy Liang. Understanding black-box predictions via influence functions.
> [2] 	Ali Shafahi, W. Ronny Huang, Mahyar Najibi, Octavian Suciu, Christoph Studer, Tudor Dumitras, and Tom Goldstein. Poison frogs! targeted clean-label poisoning attacks on neural networks.
> [3]	Jacob Steinhardt, Pang Wei Koh, and Percy Liang. Certified defenses for data poisoning attacks.
> [4]	Andrea Paudice, Luis Munoz-Gonzalez, and Emil C. Lupu. Label sanitization against label flipping poisoning attacks.
> [5] 	Pang Wei Koh, Jacob Steinhardt, and Percy Liang. Stronger data poisoning attacks break data sanitization defenses.

---

> ### Author Response · Authors · 2019-11-14
> **We have updated our paper**
>
> We have updated our paper to address your concerns, making the abstract more explicit about the use of linear classification and discussing related work with respect to regression. Please let us know if there are any further clarifications we can make!

---

### Decision · Program_Chairs · 2019-12-19

**Decision:**

Reject

**Comment:**

The authors develop a certified defense for label-flipping attacks (where an adversary can flip labels of a small number of training set samples) based on the randomized smoothing technique developed for certified defenses to adversarial perturbations of the input. The framework applies to least-squares classifiers acting on pretrained features learned by a deep network. The authors show that the resulting framework can obtain significant improvements in certified accuracy against targeted label flipping attacks for each test example.

While the paper makes some interesting contributions, the reviewers had the following shared concerns regarding the paper:
1) Reality of threat model: The threat model assumes that the adversary has access to the model and all of the training data (so as to choose which labels to flip), which is very unlikely in practice.
2) Limitation to least squares on pre-trained features: The only practical instantiation of the framework presented in the paper is on least squares classifiers acting on pre-trained features learned by a deep network.

In the rebuttal phase, the authors clarified some of the more minor concerns raised by the reviewers, but the above concerns remained.

Overall, I feel that this paper is borderline - If the authors extend the applicability of the framework (for example relaxing the restriction on pre-training the deep features) and motivating the threat model more strongly, this could be an interesting paper.